# Bedside Evaluation of Early VAS/NRS Based Protocols for Intravenous Morphine in the Emergency Department: Reasons for Poor Follow-Up and Targeted Practices

**DOI:** 10.3390/jcm10215089

**Published:** 2021-10-29

**Authors:** Virginie Eve Lvovschi, Karl Hermann, Frédéric Lapostolle, Luc-Marie Joly, Marie-Pierre Tavolacci

**Affiliations:** 1Emergency Department, UNIROUEN, INSERM U 1073, Rouen University Hospital, INSERM CIC-CRB 1404, F-76031 Rouen, France; 2Rouen University Hospital, INSERM CIC-CRB 1404, F-76000 Rouen, France; bareherrmann.karl@gmail.com; 3SAMU 93-UF Recherche-Enseignement-Qualité, Université Paris 13, Sorbonne Paris Cité, INSERM U 942, Hôpital Avicenne, F-93009 Bobigny, France; frederic.lapostolle@avc.aphp.fr; 4Emergency Department, Rouen University Hospital, F-76031 Rouen, France; Luc-Marie.Joly@chu-rouen.fr; 5Normandie University, UNIROUEN, INSERM U 1073, Rouen University Hospital, INSERM CIC-CRB 1404, F-76031 Rouen, France; MP.Tavolacci@chu-rouen.fr

**Keywords:** severe pain, oligoanalgesia, intravenous morphine titration, pain management, emergency care, opioids

## Abstract

Intravenous (IV) morphine protocols based on patient-reported scores, immediately at triage, are recommended for severe pain in Emergency Departments. However, a low follow-up is observed. Scarce data are available regarding bedside organization and pain etiologies to explain this phenomenon. The objective was the real-time observation of motivations and operational barriers leading to morphine avoidance. In a single French hospital, 164 adults with severe pain at triage were included in a cross-sectional study of the prevalence of IV morphine titration; caregivers were interviewed by real-time questionnaires on “real” reasons for protocol avoidance or failure. IV morphine prevalence was 6.1%, prescription avoidance was mainly linked to “Pain reassessment” (61.0%) and/or “alternative treatment prioritization” (49.3%). To further evaluate the organizational impact on prescription decisions, a parallel assessment of “simulated” prescription conditions was simultaneously performed for 98/164 patients; there were 18 titration decisions (18.3%). Treatment prioritization was a decision driver in the same proportion, while non-eligibility for morphine was more frequently cited (40.6% *p* = 0.001), with higher concerns about pain etiologies. Anticipation of organizational constraints cannot be excluded. In conclusion, IV morphine prescription is rarely based on first pain scores. Triage assessment is used for screening by bedside physicians, who prefer targeted practices to automatic protocols.

## 1. Introduction

The management of severe acute pain in emergency departments (EDs) is problematic worldwide, as emphasized by reports regarding oligoanalgesia [1,2,3,4]. The underprescription of opioids is a major concern even if health policies have been adopted because of the opioid crisis [5,6,7], and risk factors for opioid misuse have been identified [8,9,10].

To explain this underprescription phenomenon, most authors have focused on pain evaluation failure [11,12,13,14,15,16], including two issues that influence opioid management: heterogeneous caregiver education [17,18,19,20,21,22,23] and ED crowding [24,25,26]. To circumvent these problems and enhance the homogeneity of practices, some authors have encouraged organizational responses, supported by standardized visual analog scale (VAS) or numeric rating scale (NRS) triage-based protocols [27,28,29,30,31,32,33] as opioid initiation by a nurse [34,35]. In France, intravenous (IV) morphine, titrated according to pain assessment scores, is the “gold standard” [36]. Its safety, feasibility, and efficiency have been demonstrated in large studies [37,38]. Large and early protocols have been recommended since 2010 for patients with severe pain identified at triage before clinical examination by physicians and diagnostic hypothesis, solely triggered by self-assessment, whatever the etiology (VAS ≥ 60/100 or NRS ≥ 6/10. However, severe pain management in EDs remains critical: low rates of morphine prescription and delays before IV administration [39,40,41] are indicative of the challenges concerning physician compliance with health policies [42,43,44,45].

The practice experience of physicians and nurses is known to influence the daily use of morphine protocols in the ED. Qualitative evaluations have been performed focused on cognitive components [46,47,48,49,50]; however, there remains a lack of quantitative data regarding the ED organizational impact on eligibility for IV morphine titration at the bedside, as well as its impact on protocol deviations. The need for venous access and a nurse exclusively dedicated to titration (high level of resources) could be organizational barriers, leading to therapeutic alternatives, more available, resulting in a morphine-saving strategy, especially in case of ED crowding. Other routes of administration could be interesting alternatives [51,52,53], still, so far, underprescription of IV morphine protocols has been poorly investigated regarding routine organizational constraints in the ED (e.g., logistics and nurse availability). Moreover, the state of knowledge in the field of opioid management has changed considerably since 2010. Physicians are more aware of the variability of opioid response according to pain etiology, as well as the advantages of analgesic associations (multimodality approach) [54,55,56,57]. Some non-opioid alternatives have been validated as first-line management for severe acute pain in specific pain patterns: uncomplicated renal colic, musculoskeletal disease, headache, and neuralgia [58,59,60]. Nevertheless, there remains minimal evidence regarding the impacts of these changes on individual prescribing behavior.

We conducted a real-time survey of ED staff compliance with recommendations and barriers to early IV morphine titration from its prescription to its administration. We evaluated the IV morphine prevalence and motivations for protocol avoidance at bedside within a population that exhibited severe pain at triage, according to systematic self-assessment. Moreover, to isolate the impact of the ED organization on prescription decisions, we used a specific design with a standard epidemiological approach, taking into account the ED multiprofessional context, to assess the decision-making process [61] leading to IV morphine prescription.

## 2. Materials and Methods

### 2.1. Setting

We conducted a single-center study in the ED of the French Rouen University Hospital. This large ED (110,000 visits per year) is divided into one triage area, one critical care area, and six areas for non-critical care (medical or surgical), with two ambulatory pathways. VAS/NRS triage protocols have been implemented since 2015. VAS/NRS assessments are systematically performed in the triage area of our ED and compulsorily recorded in the electronic health records (M-UrQual software v. 7) of our hospital’s information system (HEO software 8.2; v 8.2; Maincare Solutions, France). If the pain assessment is not recorded, the remaining medical records cannot be completed by the triage nurse.

VAS/NRS assessment is compulsory even when patients do not spontaneously express pain. IV morphine titration is a daily practice recommended as an automatic analgesic modality (local guidelines), isolated or combined with other methods and/or drugs for each patient with severe pain identified at triage and admitted in the care areas, regardless of the time of day.

### 2.2. Design

We performed a 1-month evaluation of professional practices. IV morphine titrations were prospectively assessed, with the real-time investigation of factors that could influence the decision for protocol prescription; final morphine delivery was also assessed. We focused on the physician decision-making process [61] leading to prescription, as well as organizational components that could become negative constraints, to assess the prescription framework of IV morphine titration. For this assessment, we combined a cross-sectional study design (following the STROBE guidelines [62]) with an innovative method involving bedside simulation of prescription decision, supported by a case-based reasoning approach [63] that was adapted for this assessment; the study flowchart is presented in Figure 1.

In the cross-sectional study, we included patients over 18 years of age who exhibited severe pain at triage (i.e., VAS ≥ 60/100 or NRS ≥ 6/10); we measured the morphine titration prevalence among these patients. From 08:00 to 17:00 on weekdays, all patients who presented in ED care areas (except critical care) were screened in real time, using electronic health records from triage and care areas.

For each included patient, separate standardized interviews were conducted with the physician and the nurse providing direct care to the patient; these interviews were used to record their real-time intentions to perform an IV morphine protocol, as well as their motivations in the event of protocol avoidance or incomplete procedures (Figure 2). The interviews were divided into three chronological steps: (1) one regarding physician decision of prescription, (2) one regarding physician confirmation of the protocol, and (3) one regarding nurse administration. At the first step, the prescriber was questioned concerning her/his prescription decision (yes/no). If the response was “no,” declared reasons for protocol avoidance were collected, and the interview was finished. If the response was “yes,” the second step was performed. If the IV morphine protocol was canceled, related reasons were collected. If the prescription was maintained, the third step of the interview related to nurse administration was required. In the event of non-administration, related reasons were collected.

Simultaneously, for each included patient, the investigators interviewed a senior physician not providing direct care to the patient, in the form of an ancillary study; this interview consisted of a parallel assessment of the decision-making process leading to IV morphine prescription. The senior physician was asked to simulate a decision concerning protocol follow-up (without patient questioning or clinical examination) solely on the basis of triage and health records; the physician also considered available organizational resources (human and logistic) in the patient care area. This physician was blinded to the real decisions of the physician and nurse providing direct care to the patient. This novel assessment approach was possible because (almost daily) one physician in our ED is detached from direct care and dedicated to organizational matters or the inclusion of patients in clinical research protocols. This is a senior physician from the team; the identity of the physician changes daily.

The rate of protocol follow-up in the simulated condition of severe pain management was measured and compared with morphine titration prevalence in the real condition of a direct physician-patient relationship. Moreover, in a one-step interview, the decision-making process leading to morphine protocol avoidance in the simulated condition was evaluated via self-assessment. Related reasons in the event of IV morphine avoidance in the simulated condition were recorded. At the end of the investigation, all data were collated and anonymized for analysis.

Furthermore, we performed a subgroup analysis involving patients with a decision for protocol avoidance in both real and simulated prescription conditions. Overlapping reasons leading to the use of IV morphine-saving strategies were identified.

### 2.3. Data Collection and Evaluation Criteria

Questionnaires included items related to situations considered as well-known factors of oligoanalgesia. Factors linked to morphine avoidance (already identified or presumably involved, according to current literature) were divided according to classical domains: patient characteristics, physician characteristics, and organizational components. Moreover, we met ED nurses several times to collect information to complete the study questionnaire with supplementary items. Four mixed panels of grade, age, gender, and responsibility level were constituted to ensure representativeness of the nurse population.

The questionnaire comprised a 60-item grid to evaluate the real intention to perform a morphine protocol (Figure 2). The first step included 37 items from 6 patterns that explored various factors leading to a morphine-saving strategy; for example, one pattern was dedicated to the need for a re-assessment of pain, with the aim of confirming an initial high score at triage. The second step included one mixed pattern of six other items related to work organization in the form of team shifts, as well as a multiprofessional revised decision. The third step included 17 items, divided into three motivation patterns and a supplementary work-load pattern.

For physicians in the simulated condition, the grid was reduced to items included in the first step. Moreover, items linked to direct clinical evaluation of the patient were removed or converted into “planned” items.

The grade (senior or junior), gender, and age of physicians were recorded, as were the following patient characteristics: pathway components (arrival time, admission route, and discharge mode) and VAS/NRS pain scores. For interpretation purposes, patients were classified according to VAS/NRS elementary intervals (10 for VAS and 1 for NRS). Patients were also classified according to their pain-related patterns on final recorded diagnosis at discharge (traumatological, visceral, and urogenital) and medical patterns (musculoskeletal, including spinal disorders, and non-musculoskeletal, including medical thoracic pain or headache).

### 2.4. Statistical Analysis

Qualitative variables (described as percentages) were compared using Fisher’s exact test. Continuous variables (described as means ± standard deviations) were compared using Student’s *t*-test. Statistical significance was defined as *p* < 0.05, and analyses were performed using XLSTAT Biomed v. 19.5 (2017).

## 3. Results

A total of 164 patients aged 18 to 96 years (mean age, 45.9 ± 20.1 years; 54.2% women) were included in this cross-sectional study in January 2019. The daily rates of inclusion (mean patient number, 8.7 ± 5.7) and of morphine protocols performed are shown in Appendix A Appendix A. Each day, zero to 20 patients were eligible for the IV morphine protocol. There were 11 days without titrations, otherwise the titration rate varied from 5% to 18%.

NRS was the only assessment tool chosen by triage nurses; the mean NRS was 7.5/10 (standard deviation = 1.3). The NRS interval proportions were: NRS 6/10 (26.7%), NRS 7/10 (26.1%), NRS 8/10 (24.2%), NRS 9/10 (11.5%), and NRS 10/10 (11.5%). The pain pattern proportions with their respective mean NRSs were: traumatological, 23.1% (mean NRS, 7.3 ± 1.3); visceral and urogenital, 19.6% (mean NRS, 7.7 ± 1.3); non-musculoskeletal, 43.7% (mean NRS, 7.5 ± 1.3); and musculoskeletal, 13.7% (mean NRS, 7.8 ± 1.3). The mean NRS did not differ according to pain pattern (*p* = 0.52).

### 3.1. IV Morphine Prevalence

The prevalence of IV morphine titration was 6.1% (10/164; 95% confidence interval, 2.4–9.8). All decided titrations were confirmed and administered. Therefore, no nurse interviews were needed. The patient age, NRS, and pain-related pattern did not differ according to physician grade (Table 1).

In the ancillary study, 11 senior physicians simulated prescription decisions for IV morphine titration in 98/164 patients; the rate of protocol follow-up was 18.3% (18/98; 95% confidence interval, 11.5–27.7). This rate was significantly higher than in the “real” condition of a direct physician-patient relationship (*p* = 0.003); it did not significantly differ among the 11 physicians (*p* = 0.06).

### 3.2. Reasons for Protocol Avoidance

Concerning declared reasons for IV morphine avoidance, in real and simulated prescription conditions, the proportions of the six possible patterns are shown in Figure 3 and compared in Table 2.

For the real prescription condition, the main reasons for protocol avoidance were linked to pain reassessment, including a high proportion of “VAS/NRS subjective reassessment” by physicians (48.7%). “Prioritization of an alternative treatment” (49.3%), was the second item most frequently recorded, and “another class of analgesic“ (35.7%), the second reason. For the simulated prescription condition, the main motivation for IV morphine avoidance was “prioritization of an alternative treatment” (45.9%), including reduced intention to use competing analgesics (30.6%); items related to “morphine non-eligibility” were more frequent (40.6% vs. 22.7% in the real condition; *p* = 0.001) and were mostly linked to pain etiology considerations. Moreover, for 20% of patients, physicians in the simulated condition declared that they would have reassessed pain levels if they had been involved in a physician-patient relationship.

### 3.3. Subgroup Analysis

We considered the 79 patients in the real and simulated prescription conditions who did not receive IV morphine. In 51.9% of patients (41/79), we found one overlapping reason for IV morphine avoidance between physicians in both conditions. Detailed data are presented in Appendix A. The item “prioritization of another class of analgesic” was the most frequent same reason (20 times); in the other cases, the overlap was linked to pain-related patterns. Discordant decisions appeared more frequent in patients with traumatological presentations, but diverse reasons were provided.

## 4. Discussion

Our findings confirmed the daily use of morphine-saving strategies in the ED [64,65]; they highlighted the poor applicability of automatic protocols for using IV morphine at triage.

First, we observed a low rate of IV morphine titrations (prevalence < 10%), despite exhaustive patient self-assessment via mandatory VAS/NRS assessments using a computerized control system. Underassessment is regularly suggested as a major contributor to morphine undertreatment in the ED. Thus, our result challenges the findings in prior literature. Second, we observed no operational constraints that comprised organizational barriers to IV morphine administration. Poor morphine administration was the only reflection of the low rate of prescription decisions. All prescriptions of morphine protocols were administered. Poor protocol compliance was not caused by reluctance to use the IV titration method (no oral opioid or IV morphine was prescribed as an alternative). Moreover, no conflict with test scheduling during the ED visit was observed; IV morphine avoidance linked to the risk of “transfer to the operating room/department/tests within 30 min” was rare (<1.5%).

Third, considering prescribing behaviors, our results suggest that avoidance of morphine titration was not linked to a lack of knowledge regarding adequate morphine indications, questioning the relevance of pain programs that are mainly focused on this concern [66,67]. Physicians appeared to need time to contextualize their initial opioid prescription in the care area; they balanced initial nurse pain assessment at triage (by a single VAS/NRS) with other patient assessments, after a complete patient evaluation, as it is recommended for pain management outside the EDs. This study shows that even in the Emergency Department, IV morphine prescription is a multi-criteria decision-making process. A modern approach with prioritization of pain patterns and etiological treatments to guide pharmacological choice was highlighted. Our results confirm the difficulties of measuring operational components involved in individual therapeutic decisions. Despite the study design, we failed to find a direct relationship with operational components. Under the simulated condition of prescription, cognitive components and organizational motivations were voluntarily weighted for comparison with factors linked to the physician-patient relationship. The high rate of protocol avoidance (>80%), as indicated by the significantly higher rate of “morphine non-eligibility” in the simulated condition, reinforced the hypothesis of pain-pattern considerations as a priority for ED practitioners. Nevertheless, operational components, expected to be masked, were not emphasized.

### 4.1. Relevance of First NRS/VAS Assessment at Triage

We observed physician reluctance to use the first patient-reported VAS/NRS assessment at triage, as a sufficient trigger for guiding analgesic choice. “Subjective NRS/VAS reassessment” and “prioritization of another class of analgesics” were the two main items found in this study: these two methods for protocol avoidance correspond to the same choice. The first VAS/NRS assessment at triage was also questioned by 20% of physicians in the simulated condition. This result reinforces prior questions regarding the relevance of early systematic patient-reported scoring [22,68,69,70,71,72]. To contextualize a patient’s pain assessment in the ED, all interactions between the patient and his/her environment must be integrated with respect to their evolution during the ED visit. The patient must manage successive exchanges, expressing pain to different care providers in different care areas, rather than a one-way narrative process. Communication is motivated by a therapeutic alliance with goals of consideration, pain relief, and satisfaction. In accordance with an implicit contract, patients must guess the physician’s or nurse’s scoring expectations at each assessment [73]; a reciprocal interaction must also be considered [74,75]. Moreover, the use of mandatory VAS/NRS assessments at triage could lead to overassessment. Our study likely included patients who usually tolerate their pain, rather than express it spontaneously. Pain management based on patient request for pain medication has been proposed as an alternative [76,77].

Subjective physician downgrading of VAS/NRS findings is problematic. The lack of adequate tools to describe the therapeutic alliance in the medical records and criteria for pharmacologic choices cannot serve as an excuse for misusing patient-reported pain scores. VAS/NRS tools were designed to reliably measure pain sensation from a patient’s perspective [78,79,80]; when caregivers consider these patient-reported scores in the context of their own subjectivity, the validity of these tools is lost. Moreover, in this type of hidden third-party assessment, no alternative reliable pain measure is proposed. VAS/NRS physician interpretation cannot be useful for repeated evaluations, especially in a multiprofessional context leading to shared pain management. In our study, the rate of “rescue” pain assessment by the Simplified Cognitive Scale was low (≤1%). Therefore, to complete the approach with patient-reported scoring, further investigations are required to evaluate the expectations of physicians regarding objective pain assessment tools.

### 4.2. Prioritization of Pain Patterns

Our results indicate that a one-size-fits-all protocol (regardless of the pain mechanism), is no longer clinically relevant. In this study, the prioritization of alternative treatments led to a significant morphine saving. Opiophobia was not found as a decision driver for this saving strategy. Very rare concerns about drug addiction or morphine dependence were reported by physicians (≤1%). First-line treatments were chosen to prioritize etiological treatment; and the high proportion of “morphine non-eligibility” reasons suggests that although large protocols may have been a logical first step to implement pain protocols in the oligoanalgesia context, they are now outdated. Nearly 50% of patients were considered by physicians to be outside the scope of the recommended IV morphine protocol on the basis of pain typology. This result was particularly obvious in the ancillary study, including the subgroup analysis. Emergency physician practices might be consistent with complementary learned society recommendations in particular specialties [58,60], all of which were established after 2010. Currently, opioids are not indicated in specific pain patterns, even in patients with severe pain (e.g., renal colic and minor musculoskeletal pathology), sometimes causing hyperalgesia (as in cephalalgia); IV morphine protocols would represent misuse in these patients. Thus, further investigations are required to evaluate the impact of educational level on this prescription rationalization according to pain pattern.

### 4.3. Reflex of Organizational Fitness

Intravenous morphine titration is more nurse time-consuming than other orally administered analgesics, and the availability of examination rooms is variable, sometimes with long delays. In ambulatory trauma care, inhaled analgesics are proposed to shorten the time to pain relief; they offer an intriguing alternative to the organizational constraints of IV morphine titration in ambulatory pathways [81]. Our results might suggest that prescription decision for IV morphine titration does not depend on organizational components. However, VAS/NRS misappropriation and the prioritization of pain patterns could be explained by concerns about organizational fitness. This concept, described in other research areas [82,83], is related to the efficiency anticipation that leads to an ability to adapt in a care environment. In this study, physicians might have unconsciously anticipated organizational difficulties linked to nurse availability as a reflex. Care environment-related concerns are probably undeclared or integrated co-factors of morphine-saving strategies. This interpretation is consistent with studies regarding the adjustment of patient-reported scores at ED triage [46,73]: subjective downgrading or subjective upgrading could be used to set care priorities. In this study, this phenomenon could be observed regardless of crowding. Indeed, retaining adherence to a protocol consisting of automatic IV titration for each eligible patient can represent an unrealistic goal even in standard occupancy: Extreme variability of the number of eligible patients per day could affect compliance (cf Appendix A). This aspect of organizational fitness is difficult to isolate in classical analyses of the prescription framework; it may require other evaluation methods [84]. Multi-criteria decision analysis models can be used to evaluate complex decision-making processes that involve organizational components [85], including pain management [86].

### 4.4. Limitations

The main limitation of this study was its single-center setting, with a weekday design, which might have introduced selection bias. The results might have been different in a non-teaching hospital that lacked pain experts. Nevertheless, our medical and paramedical staff is large and representative of ED caregivers. This study was conducted in a sample of a typical population eligible for IV morphine protocols in daily routine.

With respect to our methodological choices, other limitations are worthy of discussion. First, our bedside approach focused on anonymized reasons for protocol avoidance, and the individual caregiver data were not exhaustive. Second, there was no quantitative evaluation of individual physician workload, but the global activity indexes of our ED show standard occupancy. Furthermore, the total numbers of physicians and nurses per care area were similar to expected values, thus avoiding bias linked to unusual crowding [25]. Third, this study focused on caregivers, rather than patients; no real-time data concerning patient desire for analgesics [69,87] or past history of pain-related treatment were available. Impact on pain relief of the prescription decisions has not been investigated, epidemiological risk factors for undertreatment were not evaluated. Finally, this study lacked complementary patient assessment by DN4 score or psychobehavioral or anxiety questionnaires [88,89,90].

## 5. Conclusions

Contrary to what was expected, organizational difficulties, supposed as barriers for the realization of IV morphine titration, were not reported as direct reasons for morphine avoidance. Poor protocol compliance could lie upstream of feasibility concerns. The advance prescription of morphine, based solely on the assessment of pain intensity at triage, using current VAS/NRS assessment, lacks physicians’ adherence, without the obvious influence of the administration route of morphine. This study assessed the weight of the individual therapeutic alliance, based on the direct physician-patient relationship, despite pressure for standardization and nurse delegation. Rather than underprescription, this study underlined a targeted practice, in which pain patterns were prioritized as well as pharmaceutical eligibility. New eligibility criteria for morphine protocols should be adapted to the complexity of this prescription framework to avoid VAS/NRS misuse. Evaluation of treatment requirements and consideration of problematics as organizational fitness should become usual approaches to limit oligo-analgesia in the ED. Some complementary trials focusing on patient relief are needed so that the guidelines could integrate these data from health-workers observation.

## Figures and Tables

**Figure 1 jcm-10-05089-f001:**
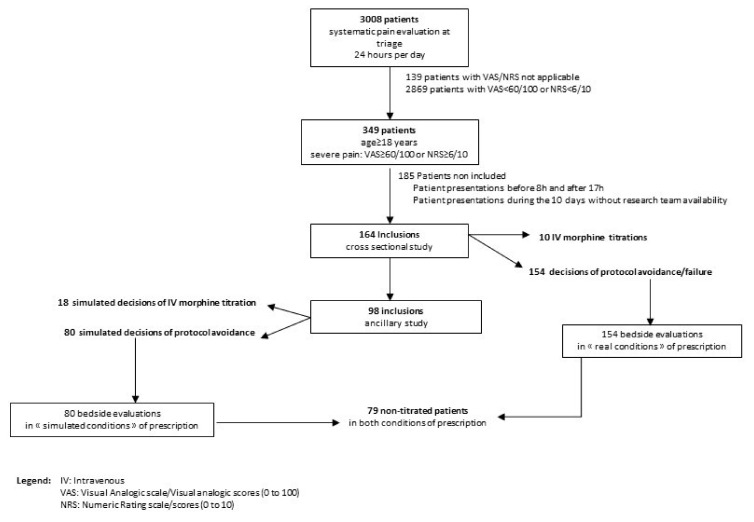
Study flowchart.

**Figure 2 jcm-10-05089-f002:**
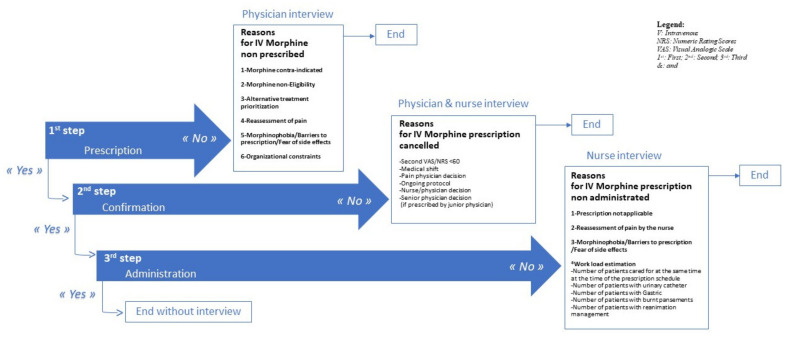
The 1 to 3-step caregivers’ interview: patterns and items.

**Figure 3 jcm-10-05089-f003:**
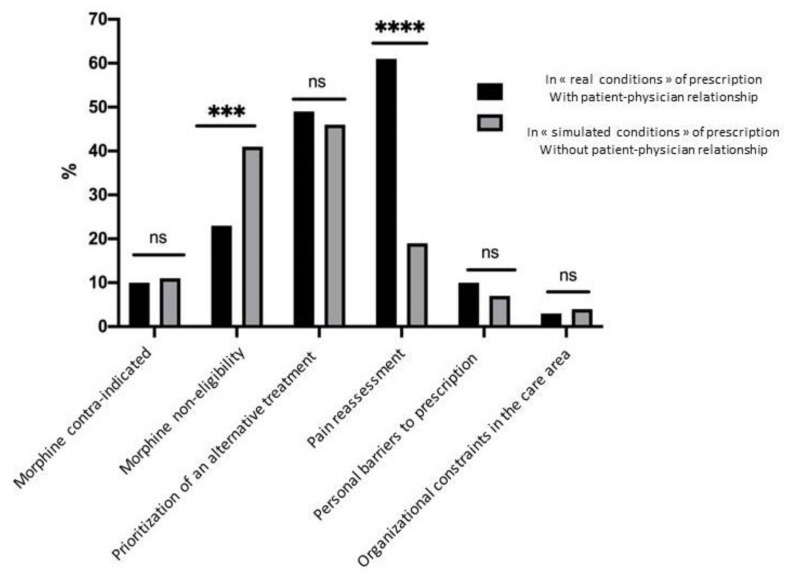
Patterns of prescription leading to protocol avoidance, comparison in both conditions of prescription: « real » vs. « simulated ». *** means *p* ≤ 0.001, ****means *p* ≤ 0.0001.

**Table 1 jcm-10-05089-t001:** Characteristics of patients in the “real” condition of prescription according to physician grade (*n* = 164).

	Junior Physician (*n* = 92 Patients)	Senior Physician (*n* = 72 Patients)	*p*
Patients’ age (years), mean (SD)	45.9 (20.9)	45.9 (19.2)	0.77
NRS, mean (SD)	7.40 (1.4)	7.7 (1.9)	0.10
Pain-related patterns (%)			0.74
- Traumatological	20.6	24.6	
- Visceral and urogenital	15.2	13.7	
- Non-musculoskeletal	54.4	48.0	
- Musculoskeletal	9.8	13.7	

SD: Standard deviation; NRS: Numeric Rating Scores.

**Table 2 jcm-10-05089-t002:** Reasons for protocol avoidance leading to morphine saving at bedside in the two parallel assessments of professional practices: in the “real” condition vs. in the “simulated” condition of prescription.

Reasons for Protocol Avoidance	In the “Real” Condition of Prescription (*n* = 164)%	In the “Simulated” Condition of Prescription (*n* = 98)%	*p*
Lack of communication about NRS level between nurses from ED triage to care area	0.6	0	
NRS between 6 and 7	1.2	10.2	
**Morphine contra-indicated**	**9.7**	**10.2**	**0.78**
Allergy	0.6	0.0	
Patient’s background	8.4	8.2	
Vital signs	0.6	2.0	
Drowsiness	0.0	1.0	
**Morphine non-eligibility**	**23.3**	**40.6**	**0.001**
*Pain-related patterns*	22.7	40.6	
Headache	6.5	12.2	
Lumbago, rachialgia, neuropathy	3.3	5.1	
Minor trauma	8.4	12.2	
Expected fracture or dislocation treatment in the ED	0.6	4.1	
Renal colic	3.9	7.1	
Pharmacokinetic concerns: First dose of morphine (excluding titration) given on arrival in ED	0.6	0.0	
**Prioritization of an alternative treatment**	**49.3**	**45.9**	**0.85**
Oral opioids	0.0	0.0	
IV morphine administration without titration	0.0	0.0	
Etiological treatment (antibiotics, immobilization, antineuropathics)	13.6	15.3	
Prioritization of another class of analgesics (acetaminophene, etc.)	35.7	30.6	
**Pain reassessment**	**61.0**	**19.4**	**<0.0001**
VAS/NRS subjective reassessment by physician	48.7	14.3	
Second VAS/NRS pain assessment by the patient at the physician’s request	11.7	4.1	
Pain assessment with a simplified cognitive scale by physician	0.6	1.0	
**Personal barriers to prescription**	**9.7**	**7.0**	**0.56**
Physician does not wish to use venous route	0.6	0.0	
Patient with unpleasant/aggressive behavior	1.2	0.0	
Patient refusal of another class of analgesic	3.5	0.0	
Fear of ineffectiveness with low standard doses	0.0	0.0	
Fear of patient dependence	0.6	0.0	
Patient with advanced age	1.2	1.0	
Patient with drug addiction	0.0	1.0	
Patient with alcoholism	0.0	0.0	
Fear of sedative drug combination	1.9	1.0	
Fear of occlusive syndrome (possible increased vomiting)	0.6	2.0	
Other respiratory depressants already prescribed	0.0	2.0	
**Organizational constraints in the care area**	**2.5**	**4.0**	**0.48**
No scope available	0	0.0	
Transfer to the operating room/department/tests within 30 min	1.3	0.0	
No booth available within 30 min to initiate the prescription	0	1.0	
Morphine not available	0	0.0	
Lack of nurse availability to initiate titration/understaffing	0.6	1.0	
Titration monitoring not feasible every 5 min	0.6	1.0	
Lack of nurse availability for monitoring	0.0	1.0	

The titles of the patterns, provided in the interview grid, are in bold type. The thematic clusters that emerged from the interviews are in italics. NRS: Numeric Rating Scores, ED: Emergency Department.

## Data Availability

The data presented in this study are available on request from the corresponding author. The data are not publicly available due to health records utilization.

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
