# Peer review of "Bedside Evaluation of Early VAS/NRS Based Protocols for Intravenous Morphine in the Emergency Department: Reasons for Poor Follow-Up and Targeted Practices"

_jcm, 2021, doi:10.3390/jcm10215089_

Round 1

Reviewer 1 Report

The manuscript by   Lvovschi  et al aims to gain understanding about organizational barriers and decision making among ER physicians  leading to compliance  with guidelines for administering IV opioids to patients reporting severe pain and admitted to the triage area (ie not critical cases). 

The study consisted of two phases: (1) interviewing physicians for reasons of complying  (or not) with the French guidelines for  treating patients reporting severe pain in the ER ; (2) presenting the clinical findings to the senior physician in charge of the ER at the time to simulate the decision-making process and evaluate whether similar reasons would be reached for treating this particular patient.

This is a single center study, in a large ER in France.   

A  strength of the study was interviewing  practitioners in real time rather than relying  retrospective interview and detailed analysis of the reasons for administering IV opioids to triage patients.

Readability of the  current version is very much improved compared to an earlier version of this manuscript; the authors carry out an exhaustive evaluation of the reasons for  inadequate treatment of pain, and exhaustive explanation of their findings. Consequently the text in the Introduction and Discussion tends to be long and at times, unfocused and unwieldy.

The findings from the study by Lvovoschi et al  indicate that administering IV morphine in more nuanced than automatic administration of morphine.  I doubt opioids are administered automatically in any ER.  It seems that this is the main message of this study and is stated in the conclusions to the study. Indeed the authors call for revision & updating  of the French guidelines.

The study is nicely carried out, indicating that the clinical decision making and structures in this ER  function well. However,  I do not think the findings provide clinicians with new and novel knowledge:

  • Physicians in this ER follow current practice according to which IV opioids are not administered based on one evaluation of pain,  etiology of the pain is carefully evaluated, physicians  require that additional assessments of pain are carried out; they seek  out alternative means for analgesia.
  • The authors write that  pain management is not administered in a ‘one size fits all’ fashion. Discussion,, Line 314 or that  ‘automatic protocols for using IV morphine at triage’ are not applicable. (Discussion, Lines 245-246). However, personalization of treatment with opioids or any medication/treatment  is a premise of any guideline.  The role of guidelines or local protocols is never prescriptive and clinicians should  to use their judgment with every patient. Therefore, this is not a new findings but as clinical practice should be.

Abstract

Line 17:  The authors write   ‘low follow-up’. Do they intend to write that  ‘adherence’ to this protocol is low?

Line 22-24 What do the authors mean by ‘Prevalence was 6.1% only linked to prescribing decision ’?  What do they mean by:   “Pain reassessment” (61.0%), “alternative treatment prioritization” (49.3%), “morphine non-eligibility” (23.3%) were involved 

These numbers add up to more than  100% - does it mean there were several reasons for non-administration of  IV opioids to some patients?

Introduction

Line 47-48: What do the authors mean by ‘Large and early protocols have been recommended’

What do these protocols recommend?

Line 52 – ‘poor follow up’ ?  Follow-up of what?  Or do the authors aim to address ‘implementation of guidelines’ ?

Aims of the study (lines 70-77) are clearly stated but the paragraph preceding this (lines 52-69) is long and un- focused – what do the authors intend to write?

Figure 1- Study flow chart: what does ‘EVA’ stand for ? ‘and ‘EN’.

What do the authors mean by 100/100 > VAS >60/10 and the same notation for the NRS scale.

Discussion

The conclusion (lines 371-380)  is a clear summary – though the statements do not provide novel information to clinicians, eg

IV morphine prescription is a multi-criteria decision-making process.

The first patient-reported VAS/NRS assessment at nurse triage is an insufficient prescription trigger.

Author Response

We would like to thank reviewer#1 for providing us with the opportunity to improve our manuscript. Please find below our responses to the requests and to the questions raised by reviewer#1, as well as the location of the changes made in the manuscript (clean version), particularly in the abstract, in the introduction, and in the discussion section.

We hope that our answers will meet expectations. We remain available for any further corrections.

Dr Virginie-Eve Lvovschi

General comments

The manuscript by   Lvovschi  et al aims to gain understanding about organizational barriers and decision making among ER physicians  leading to compliance  with guidelines for administering IV opioids to patients reporting severe pain and admitted to the triage area (ie not critical cases). 

The study consisted of two phases: (1) interviewing physicians for reasons of complying  (or not) with the French guidelines for  treating patients reporting severe pain in the ER ; (2) presenting the clinical findings to the senior physician in charge of the ER at the time to simulate the decision-making process and evaluate whether similar reasons would be reached for treating this particular patient.

This is a single center study, in a large ER in France.   

A  strength of the study was interviewing  practitioners in real time rather than relying  retrospective interview and detailed analysis of the reasons for administering IV opioids to triage patients.

Readability of the  current version is very much improved compared to an earlier version of this manuscript; the authors carry out an exhaustive evaluation of the reasons for  inadequate treatment of pain, and exhaustive explanation of their findings. Consequently the text in the Introduction and Discussion tends to be long and at times, unfocused and unwieldy.

The findings from the study by Lvovoschi et al  indicate that administering IV morphine in more nuanced than automatic administration of morphine.  I doubt opioids are administered automatically in any ER.  It seems that this is the main message of this study and is stated in the conclusions to the study. Indeed the authors call for revision & updating  of the French guidelines.

The study is nicely carried out, indicating that the clinical decision making and structures in this ER  function well. However,  I do not think the findings provide clinicians with new and novel knowledge:

  • Physicians in this ER follow current practice according to which IV opioids are not administered based on one evaluation of pain,  etiology of the pain is carefully evaluated, physicians  require that additional assessments of pain are carried out; they seek  out alternative means for analgesia.
  • The authors write that  pain management is not administered in a ‘one size fits all’ fashion. Discussion,, Line 314 or that  ‘automatic protocols for using IV morphine at triage’ are not applicable. (Discussion, Lines 245-246). However, personalization of treatment with opioids or any medication/treatment  is a premise of any guideline.  The role of guidelines or local protocols is never prescriptive and clinicians should  to use their judgment with every patient. Therefore, this is not a new findings but as clinical practice should be.

Response:

French recommendations and numerous reports in the international literature have been calling for the past 10 years for automatic titration with the aim of bypassing individual prescriptions. According to guidelines and many authors, the ideal pain management would be based on delegated nurse-protocols at the time of the very first evaluation in the emergency room, i.e. before any initial medical contact, especially in case of crowding. This argument is summarized in the introduction section of our manuscript. In addition, we have recalled the particular challenge of harmonization that drives health policy in the ED environment (for example line 43).

The originality of this study is to show that what is considered to be a relevant response on an organizational level is in fact not. The rate of compliance with these early protocols is known to be low, even in trained and conscious services. This study shows that this shortcoming is not simply related to the mode of administration of morphine that deserves to be modernize.

Simplifying access to morphine, does not seem to us to be an adequate response to the oligoanalgesia problem in the ED. A management of pain with morphine that finds a better balance between standardization and individualization seems to be sought by physicians. This point of view was already defended by our team in a letter published last year (doi: 10.1016/j.annemergmed.2019.08.448.).

We have tried in this new version of the manuscript to better underline the originality of our findings and of our position in the current context of literature and guidelines. Moreover, we have clarified in the introduction section the concept of “large and early protocol” and add developments lines 48-51 to facilitate the reading of the discussion line 307. The distinction between relevance and applicability is easier to understand in the new version of the manuscript: a more precise argument is proposed in the discussion section lines 341-344.

Abstract

Line 17:  The authors write   ‘low follow-up’. Do they intend to write that  ‘adherence’ to this protocol is low?

Response:

Adherence is a hypothesis, a mechanism that can be involved in poor follow-up of prescription instructions. The objective of the study was to determine whether organizational factors truly affect the practical application of protocol already endorsed by a team aware of the oligoanalgesia problematic. We therefore remained deliberately descriptive at this stage of the abstract, dedicated to current context. The low rate of prescription and completion of morphine protocols, is described in literature since 10 years (ref [40, 41] in the new manuscript).

Line 22-24 What do the authors mean by ‘Prevalence was 6.1% only linked to prescribing decision ’?  What do they mean by:   “Pain reassessment” (61.0%), “alternative treatment prioritization” (49.3%), “morphine non-eligibility” (23.3%) were involved 

These numbers add up to more than  100% - does it mean there were several reasons for non-administration of  IV opioids to some patients?

Response:

The very short format of the abstract (200 words) does not allow much elaboration, so we rather propose a reorganization and a simplification of this paragraph in the abstract, integrating the fact that several reasons for non-titration were mentioned for the same patient by some physicians:

Lines 22-24, the new sentence is: “IV morphine prevalence was 6.1%, prescription avoidance was mainly linked to Pain reassessment” (61.0%) and/or “alternative treatment prioritization” (49.3%).”

Introduction

Line 47-48: What do the authors mean by ‘Large and early protocols have been recommended’

What do these protocols recommend?

Response:

Guidelines and literature recommend “One for all” protocols, as soon as possible at the entrance of the ED (nurse-triage area), without a complete evaluation of the patient, before clinical evaluation by a physician. For concision, we have added a short explanation in the introduction, and more developments are now available in the method section.

Lines 48-51 the new sentence is “for patients with severe pain identified at triage before clinical examination by physicians and diagnostic hypothesis, solely triggered by self-assessment, whatever the etiology (VAS ≥ 60/100 or NRS ≥ 6/10).”

Lines 90-93 the new sentence is: “IV morphine titration is a daily practice recommended as an automatic analgesic modality (local guidelines), isolated or combined with other methods and/or drugs for each patient with severe pain identified at triage and admitted in the care areas….”

Line 52 – ‘poor follow up’ ?  Follow-up of what?  Or do the authors aim to address ‘implementation of guidelines’ ?

Response:

We have clarified this term, lines 54-55 the new sentence is: “Practice experience of physicians and nurses is known to influence daily use of morphine protocols in the ED.”

Aims of the study (lines 70-77) are clearly stated but the paragraph preceding this (lines 52-69) is long and un- focused – what do the authors intend to write?

Response:

As requested, in this part of the introduction section, we have simplified and focused the text on missing data, we have reduced the text to clarify the scientific context of the study (lines 54-67).

Figure 1- Study flow chart: what does ‘EVA’ stand for ? ‘and ‘EN’.

Response:

We apologize for this error of translation, we thought we had already corrected it, we have replaced EVA by VAS and EN by NRS in the updated version.

What do the authors mean by 100/100 > VAS >60/10 and the same notation for the NRS scale.

Response: we have corrected figure 1 to address this point.

Discussion

The conclusion (lines 371-380)  is a clear summary – though the statements do not provide novel information to clinicians, eg

IV morphine prescription is a multi-criteria decision-making process.

The first patient-reported VAS/NRS assessment at nurse triage is an insufficient prescription trigger.

Response:

In agreement with reviewer#1's comment, we have developed the conclusion section to address this weakness of the manuscript, in order to underline the unexpected results we observed and their relevance in the current context of promoting any advanced protocols.

Reviewer 2 Report

General comments:

In this study, the authors tried to find discrepancies between VAS/NRS protocol for intravenous morphine and realistic prescription in the emergency department. I think this discrepancy is very important and the results sound to be interesting. However, the manuscript is hard to read because of redundancy and the references are too much to follow. I think this manuscript should be thoroughly rewritten to improve readability.

Specific comments:

Abstract

P1L17: I'm not sure what the follow-up is in this sentence. It should be clearly stated.

P1L21: Is the “real-time questionnaires” for healthcare professionals?

Introduction

This section is well-written, but there are too many references than necessary.

Materials and methods

This section is well-written. However, in Figure 2, it is better to adopt a tree diagram to help the reader understand the overall research design.

Results

Figure 3 is unnecessary and should be deleted.

Discussions

More detailed consideration is needed as to why morphine administration is avoided. For example, it is desirable to have epidemiological results on the proportion of patients visiting the emergency department who are addicted to morphine.

What was the impact of morphine disqualification on the success rate of pain relief in patients, and what are the benefits to patient treatment?

Author Response

We would like to thank reviewer#2 for providing us with the opportunity to improve our manuscript. Please find below our responses to the requests and to the questions raised by reviewer#2, as well as the location of the changes made in the manuscript (clean version), particularly in the abstract, in the introduction, and in the discussion section.

We hope that our answers will meet expectations. We remain available for any further corrections.

Dr Virginie-Eve Lvovschi

General comments

In this study, the authors tried to find discrepancies between VAS/NRS protocol for intravenous morphine and realistic prescription in the emergency department. I think this discrepancy is very important and the results sound to be interesting. However, the manuscript is hard to read because of redundancy and the references are too much to follow. I think this manuscript should be thoroughly rewritten to improve readability.

Response:

We would like to thank reviewer#2 for his positive feedback on this study. We have removed redundancies that could hinder the reading, especially in the discussion, and we have made the introduction more focused. Throughout the manuscript, some references have been removed (90 vs 96) and most were reordered to facilitate the reading.

Specific comments:

Abstract

P1L17: I'm not sure what the follow-up is in this sentence. It should be clearly stated.

Response:

The format of the abstract (200 words) does not allow much elaboration (nor to attach a reference), to address this point directly in the manuscript. The term “Low follow-up” is a classical term in literature on oligoanalgesia, meaning both “low rate of prescription and completion” of morphine protocols based on titration (repeated boluses). This phenomenon is described in literature since 10 years (ref [37, 40, 41] in the new manuscript). Moreover, the manuscript have been checked by two different English consultants, confirming this statement. The terms “adherence” and “compliance” are also used in the manuscript but are reserved to the discussion section, voluntarily, referring to hypotheses leading to this low follow-up. 

P1L21: Is the “real-time questionnaires” for healthcare professionals?

Response:

Thank you for this relevant remark, this crucial information was obviously missing and we have added this information line 21.

In order to comply with the guidelines for authors (wordcount), the rest of the abstract was slightly amended.

Introduction

This section is well-written, but there are too many references than necessary.

 Response: As previously mentioned, we have removed several references.

Materials and methods

This section is well-written. However, in Figure 2, it is better to adopt a tree diagram to help the reader understand the overall research design.

Response: We have updated figure 2 to address this point.

Results

Figure 3 is unnecessary and should be deleted.

Response: We have deleted figure 3 in the body of the text and replaced it by a results description lines 186-188, these results are discussed in the discussion section lines 341-344. The figure can be now viewed in supplementary material S2.

Discussions

More detailed consideration is needed as to why morphine administration is avoided. For example, it is desirable to have epidemiological results on the proportion of patients visiting the emergency department who are addicted to morphine.

Response:

We have placed our approach at an organizational level. The objective of this study was to evaluate the reasons declared by caregivers which could hinder the rapid prescription of intravenous morphine titration in the emergency department (ED). In the limitations paragraph, we remind readers that this study focuses on healthcare professionals (lines 361-364), not on patients’ characteristics, including their expectations, and the impact in terms of analgesia of prescription decisions (no analysis of risks factors associated with undertreatment). The lack of epidemiological data is explained by the design of the study. We have added a sentence in the limitations paragraph, to reinforce this point in particular lines 362-363.

Moreover, in this study no physician mentioned the item "patient with drug addiction" as a reason for not prescribing a titration. Some developments regarding this result are now proposed lines 309-311 in the discussion section of the manuscript to deal with opiophobia concern as best as possible.

What was the impact of morphine disqualification on the success rate of pain relief in patients, and what are the benefits to patient treatment?

Response:

As previously mentioned, we performed a professional practice evaluation, that wasn’t designed to address this issue. Other studies need to be run, targeting this purpose. Patient-centered studies in terms of endpoint, comparing two types of analgesic strategy in the ED should be designed to answer this question and to complete our findings. The conclusion section has been rewritten, integrating a sentence reiterating this argument lines 381-383.

An ED management of pain with morphine that finds a better balance between standardization and individualization seems to be sought by physicians. The benefit of narrowing the spectrum of morphine prescribing in the ED is likely to be the prevention of misuse in targeted populations as it is precised line 322 of the discussion section, and the limitation of crude organizational responses (we have focused on this concern in a letter published last year, doi: 10.1016/j.annemergmed.2019.08.448.). Morphine disqualification is also an imperative in some patterns (cephalalgia) to prevent hyperalgesia. This point is now clarified line 321 of the discussion section.

This manuscript is a resubmission of an earlier submission. The following is a list of the peer review reports and author responses from that submission.

Round 1

Reviewer 1 Report

The manuscript by   Lvovschi  et al aims to gain understanding about reasons for under- treatment of severe pain with IV-opioids  in the Emergency Room (ER). The authors focus on organizational barriers and decision making among ER physicians and nurses which lead  to prescription and administration (or not) of IV opioids for patients reporting severe pain.

The study consisted of two phases: (1) interviewing physicians and nurses for reasons of treating (or not) patients reporting severe pain  with IV morphine; (2) presenting the clinical findings to the senior physician in charge of the ER at the time to simulate the decision-making process and evaluate whether a similar evaluation would be reached for treating this particular patient.

The difference between the actual treatment vs simulated (real vs expected) was 6.1% vs 18.3%.

This is a single center study, in a large ER in France.   

A  strength of the study was interviewing  practitioners in real time rather than relying  retrospective interview.

The study and findings are interesting and relevant.  My main concern is that  I found myself struggling with understanding the text. While the study is supported by an extensive literature review, and interesting to read,  there are many phrases I did  not understand – for example:

Abstract:

‘Early morphine protocols’.

‘ However, a low follow-up is observed without data  about bedside organizational aspects and pain etiology concerns in daily practice’

‘indirect morphine saving’.

 prescribing rationalization.

 organizational fitness concerns.

‘ prioritization”

… adjust first patient-reported scores and prefer targeted practices, 30 integrating pain etiology concerns.

Body of the paper:

Line 74: exhaustive population of patients

Line 106: ‘informatics data’

Line 148: ‘Questionnaires had to include large oligoanalgesia determinisms’ Line 156:  ‘morphine saving ‘ . The concept is used repeatedly

Line 240: early and large IV morphine protocols

Lines 266-268: A reflex of organizational fitness might have attenuated the direct expression of operational concerns, despite the special design we used in the ancillary study.

Line 297: ‘misappropriation‘

Line 298: therapeutic alliance

Line 307: ‘complete ‘ do the authors mean ‘complement’?

Line  362 :  ‘overcrowding’

Methods:

Line 85: Could the authors elaborate as to how the  24-hour computerized-control system for assessing pain works?

Line 86:  What do the authors mean when they write that ‘IV morphine titration is a daily practice recommended in the care areas for patients with severe pain identified at triage’.  Is this recommendation written as part of a local ER guideline?

Figure 1- Study flow chart: what does ‘EVA’ stand for ? ‘and ‘EN’.

What do the authors mean by 100/100 > VAS >60/10 and the same notation for the NRS scale. I would think that ‘VAS >60/10’ would include all patients with a VAS reading of over 60/100. They use this format in the text of the manuscript.

Figure 2:  One reason listed for none administration of the morphine is re-assessment of the pain. Why was this carried out ? To confirm the initial high reading ?